# Extracorporeal Shock Wave Enhanced Exogenous Mitochondria into Adipose-Derived Mesenchymal Stem Cells and Further Preserved Heart Function in Rat Dilated Cardiomyopathy

**DOI:** 10.3390/biomedicines9101362

**Published:** 2021-09-30

**Authors:** Pei-Hsun Sung, Mel S. Lee, Han-Tan Chai, John Y. Chiang, Yi-Chen Li, Yi-Ching Chu, Chi-Ruei Huang, Hon-Kan Yip

**Affiliations:** 1Division of Cardiology, Department of Internal Medicine, College of Medicine, Chang Gung University, Kaohsiung Chang Gung Memorial Hospital, Kaohsiung 800, Taiwan; e12281@cgmh.org.tw (P.-H.S.); chaiht@mail.cgmh.org.tw (H.-T.C.); ryichenli@gmail.com (Y.-C.L.); yiching08@gmail.com (Y.-C.C.); starchmay33@gmail.com (C.-R.H.); 2Center for Shockwave Medicine and Tissue Engineering, Kaohsiung Chang Gung Memorial Hospital, Kaohsiung 800, Taiwan; 3Institute for Translational Research in Biomedicine, Kaohsiung Chang Gung Memorial Hospital, Kaohsiung 800, Taiwan; 4Department of Orthopedics, College of Medicine, Chang Gung University, Kaohsiung Chang Gung Memorial Hospital, Kaohsiung 800, Taiwan; mellee@cgmh.org.tw; 5Department of Computer Science and Engineering, National Sun Yat-Sen University, Kaohsiung 800, Taiwan; chiang@cse.nsysu.edu.tw; 6Department of Healthcare Administration and Medical Informatics, Kaohsiung Medical University, Kaohsiung 800, Taiwan

**Keywords:** exogenous mitochondria delivery, dilated cardiomyopathy, extracorporeal shock wave, left ventricular ejection fraction, angiogenesis

## Abstract

This study tested whether extracorporeal shock wave (ECSW) supported-exogenous mitochondria (Mito) into adipose-derived mesenchymal stem cells (ADMSCs) would preserve left-ventricular-ejection-fraction (LVEF) in doxorubicin/12 mg/kg-induced dilated cardiomyopathy (DCM) rat. Adult-male-SD rats were equally categorized into group 1 (sham-control), group 2 (DCM), group 3 (DCM + ECSW/1.5 mJ/mm^2^ for 140 shots/week × 3 times/since day 14 after DCM induction), group 4 (DCM + ECSW/1.5 mJ/mm^2^/100 shots-assisted mito delivery (500 μg) into ADMSCs/1.2 × 10^6^ cells, then implanted into LV myocardium day 14 after DCM induction) and group 5 (DCM + ECSW-assisted mito delivery into ADMSCs/1.2 × 10^6^ cells, then implanted into LV, followed by ECSW/1.5 mJ/mm^2^ for 140 shots/week × 3 times/since day 14 after DCM induction) and euthanized by day 49. Microscopic findings showed mitochondria were abundantly enhanced by ECSW into H9C2 cells. The q-PCR showed a significant increase in relative number of mitDNA in mitochondrial-transferred H9C2 cells than in control group (*p* < 0.01). The angiogenesis/angiogenesis factors (VEGF/SDF-1α/IG-F1) in HUVECs were significantly progressively increased by a stepwise-increased amount of ECSW energy (0.1/0.25/0.35 mJ/mm^2^) (all *p* < 0.001). The 49-day LVEF was highest in group 1 and significantly progressively increased from groups 2 to 5 (all *p* < 0.0001). Cardiomyocyte size/fibrosis exhibited an opposite pattern of LVEF, whereas cellular/protein levels of angiogenesis factors (VEGF/SDF-1α) in myocardium were significantly progressively increased from groups 1 to 5 (all *p* < 0.0001). The protein expressions of apoptotic/mitochondrial (cleaved-caspase-3/cleaved-PARP/mitochondrial-Bax/cytosolic-cytochrome-C), fibrotic (p-Smad3/TGF-ß), oxidative-stress (NOX-1/NOX-2) and pressure-overload/heart failure (BNP/ß-MHC) biomarkers exhibited an opposite pattern of LVEF among the five groups (all *p* < 0.0001). ECSW-assisted mitochondrial-delivery into ADMSCs plus ECSW offered an additional benefit for preserving LVEF in DCM rat.

## 1. Introduction

Dilated cardiomyopathy (DCM), a non-ischemia related heart muscle disease with architectural and functional myocardial abnormality [1,2,3,4,5], represents a heterogeneous group of diseases such as genetic inheritance, inflammatory disorders, myocarditis or toxic effects from medications, alcohol or illicit drugs [6,7]. It is a common cause of heart failure (HF) worldwide. DCM is a clinical diagnosis characterized by left ventricular (LV) or biventricular dilatation and impaired contraction in the absence of coronary artery disease, hypertension, valvular heart disease or congenital heart disease [5,8]. Clinical studies have revealed that mortality from DCM is usually from sudden death or progressive pump failure [1,4,9,10]. Regrettably, despite state-of-the-art improvements in therapeutic strategies and continuing review of guidelines for DCM in the past decades [4,9,11,12,13,14], the outcomes following onset of HF have not been substantially altered and mortality remains unacceptably high in patients with advanced DCM [4,9,12,14]. Therefore, to find a new, safe and efficacious therapeutic modality is of the utmost importance for these patients [15,16].

The mechanistic basis of LV contractile dysfunction followed by LV chamber dilatation/remodeling and evolution of overt HF has been intensively investigated. It has been attributed to upregulated inflammation, enhanced oxidative stress, generation of mitochondrial reactive oxygen species (ROS) [16,17,18,19,20], alteration in intracellular Ca^2+^ homeostasis [21] and mitochondrial dysfunction with exhaustion of energy and cellular apoptosis [18,20,22,23,24], as a consequence of accumulated extracellular fibrosis [11,16,20,23], cardiomyocyte death and scar formation [11,16,23,25]. Additionally, a more insightful mechanistic basis has suggested that oxidative stress-regulated Ca^2+^/calmodulin-dependent protein kinase II (Ca MKII) phosphorylation of cardiac ion channels has emerged as a crucial contributor to arrhythmogenesis in cardiac pathology [26], especially in those DCM myocytes. This is due to the fact that cardiac excitation-contraction coupling is a highly coordinated process controlled by CaMKII and PKA [27,28]. Thus, LV contractile dysfunction is cardinally caused by derangement or abnormality of cardiomyocyte calcium handling.

Interestingly, our previous studies have demonstrated that adipose-derived mesenchymal stem cells (ADMSCs) therapy effectively protected the heart function against ischemia-related damage mainly through suppressing inflammation, oxidative stress and immunogenicity [29,30]. Additionally, our studies have further demonstrated that extracorporeal shock wave (ECSW) therapy significantly reverses ischemia-related LV dysfunction and remodeling via angiogenesis/neovascularization [31]. Moreover, our previous work has revealed that combined ADMSCs and ECSW therapy was superior to merely ADMSCs or ECSW for improving acute ischemia-related LV dysfunction [32] and salvaging the critical limb ischemia [33]. Furthermore, our recent study has demonstrated that early intramyocardial implantation of exogenous mitochondria significantly preserved LV function in DCM rodent [34]. Besides, another previous study of ours has proved that ECSW therapy augmented mitochondrial delivery into target cells and protected against acute respiratory distress syndrome [35]. These findings [16,23,29,30,31,32,33,34,35] raise the hypothesis that ECSW-assisted exogenous mitochondrial transfer to ADMSCs, followed by implantation into the mitochondrially dysfunctional cardiomyocytes, may have therapeutic potential by attenuating cardiomyocyte apoptosis and death as well as enhancing angiogenesis, thereby, preserving the LV function.

## 2. Materials and Methods

### 2.1. Ethics

All animal procedures were approved by the Institute of Animal Care and Use Committee at Kaohsiung Chang Gung Memorial Hospital (Affidavit of Approval of Animal Use Protocol No. 2016111501) and performed in accordance with the Guide for the Care and Use of Laboratory Animals. Animals were housed in an Association for Assessment and Accreditation of Laboratory Animal Care International (AAALAC; Frederick, MD, USA)-approved animal facility in our hospital with controlled temperature and light cycles (24 °C and 12/12 light cycle).

### 2.2. In Vitro Study Using the Human Umbilical Vein Endothelial Cells (Huvecs) to Determine the Impact of Stepwise Increased Ecsw on Augmenting Angiogenesis

First, the HUVECs (BCRC H-UV001; China Chemical & Pharmaceutical Co., Ltd. Taipei, Taiwan) were cultured with M199 medium (#31100-035, Thermo Fisher Scientific Inc., Waltham, MA, USA), i.e., containing 1% ECGS (#02-102, Millipore, Burlington, MA, USA) + 0.2% Heparin (5000U) + 20% fetal bovine serum (Thermo Fisher Scientific Inc., Waltham, MA, USA) + 1% PS. They were seeded (cell amount 5 × 10^5^/flask) in 25 T, and incubated for 24 h. These cells of P3–P6 were finally utilized in this study and categorized into four groups as follows: (1) HUVECs only, (2) HUVECs + ECSW (0.1 mJ/mm^2^), (3) HUVECs + ECSW (0.25 mJ/mm^2^) and (4) HUVECs + ECSW (0.35 mJ/mm^2^).

### 2.3. Procedure and Protocol of DCM Induction by Doxorubicin and Animal Grouping

First, the HUVECs (BCRC H-UV001; China Chemical & Pharmaceutical Co., Ltd., Taipei, Taiwan) were cultured with M199 medium (#31100-035, Thermo Fisher Scientific Inc., Waltham, MA, USA), i.e., containing 1% ECGS (#02-102, Millipore, Burlington, MA, USA) + 0.2% Heparin (5000U) + 20% fetal bovine serum (Thermo Fisher Scientific Inc., Waltham, MA, USA) + 1% PS. They were seeded (cell amount 5 × 10^5^/flask) in 25 T, and incubated for 24 h. These cells of P3–P6 were finally utilized in this study and categorized into four groups as follows: (1) HUVECs only, (2) HUVECs + ECSW (0.1 mJ/mm^2^), (3) HUVECs + ECSW (0.25 mJ/mm^2^) and (4) HUVECs + ECSW (0.35 mJ/mm^2^).

### 2.4. Mito Isolation and MitoTracker Staining for Mito

The procedure and protocol for isolating liver mito from rats have been first reported by our previous study [36]. Briefly, additional 6 animals were utilized, and the gallbladder and liver were removed. The liver was minced into small pieces using scissors in a beaker surrounded by ice, and then homogenized by using a Teflon pestle. The homogenate was transferred to a 50 mL polypropylene Falcon tube and centrifuged at 600× *g* for 10 min at 4 °C. The supernatants were transferred to centrifuge tubes for centrifugation at 7000× *g* for 10 min at 4 °C. The supernatants were discarded, and the pellets were washed with 5 mL ice-cold IBc. The concentrations of mito suspensions were measured using the Biuret method. Each 10 mg of isolated mitochondria was labeled with 1 μM of MitoTracker Red CMXRos (Thermo Fisher Scientific Inc., Waltham, MA, USA) through incubation at 37 °C for 30 min. Mito administration was performed for the study animals immediately after labeling (i.e., <3 h after the isolation procedure).

### 2.5. Mito DNA Copy Number in ADMSCs

Total DNA was extracted from adipose derived mesenchymal stem cells (ADMSCs) using the DNeasy Blood and Tissue kit (Qiagen, Hilden, Germany) with proteinase K and RNase treatment, according to the manufacturer’s instructions. For the copy number of mito DNA, mito DNA (ND1-mtDNA forward and reverse primers (i.e., forward primer sequence: 5′-cTcccTATTcggAgcccTA-3′; reverse primer sequence: 5′-ATTTgTTTcTgcTAgggTTg-3′) was quantified by QuantiNOVA SYBR Green PCR assay (Qiagen, Hilden, Germany) and normalized by GAPDH DNA. Triplicate assays were performed for each sample on Step One-Plus system (Thermo Fisher Scientific Inc., Waltham, MA, USA).

### 2.6. Measurement of Left Ventricular Ejection Fraction by Echocardiography

Transthoracic echocardiography was performed in each group prior to and on days 35 and 60 after DCM induction. The procedure was performed by an animal cardiologist blinded to the experimental design using an ultrasound machine (Vevo 2100, FUJIFILM Visualsonics, Inc., Toronto, ON, Canada). M-mode standard two-dimensional (2D) left parasternal-long axis echocardiographic examinations were conducted. Left ventricular (LV) internal dimensions (end-systolic diameter (ESD) and end-diastolic diameter (EDD)) were measured at the mitral valve level of the left ventricle, according to the leading-edge method of American Society of Echocardiography using at least three consecutive cardiac cycles. LVEF was calculated as follows: LVEF (%) = ((LVEDD3-LVEDS3)/LVEDD3) × 100%.

### 2.7. Western Blot Analysis of Left Ventricular Tissues

Western blot analysis was performed as described previously [34,35,36]. Equal amounts (50 μg) of protein extracts were loaded and separated by SDS-PAGE. Separated proteins were transferred to PVDF membranes, and nonspecific sites were blocked by incubation in blocking buffer (5% nonfat dry milk in T-TBS (TBS containing 0.05% Tween 20)) overnight. The membranes were incubated with the indicated primary antibodies (mitochondrial Bax (1:1000, Abcam Inc., Cambridge, MA, USA), cleaved poly (ADP-ribose) polymerase (PARP) (1:1000, Cell Signaling Technology, Inc., Danvers, Ma, USA), caspase 3 (1:1000, Cell Signaling), NADPH oxidase (NOX)-1 (1:1500, Merck KGaA, Darmstadt, Germany), NOX-2 (1:500, Sigma), cytosolic cytochrome C (1:2000, Becton Dickinson, Franklin Lakes, NJ, USA), mitochondrial cytochrome C (1:2000, BD), vascular endothelial growth factor (VEGF) (1:1000, Abcam), basic fibroblast growth factor (bFGF) (1:1000, Abcam), insulin-like growth factor (IGF)-1 (1:1000, Abcam), beta myelin heavy chain (ß-MHC) (1:1000, Santa Cruz Biotechnology, Inc., Dallas, TX, USA), α-MHC (1:1000, Santa Cruz), CD31 (1:1000, Abcam Inc., Cambridge, MA, USA), Smad3 (1:1000, Cell Signaling) and transforming growth factor TGF-ß (1:1000, Abcam Inc., Cambridge, MA, USA) for 1 h at room temperature. Horseradish peroxidase-conjugated anti-rabbit IgG (1:2000, Cell Signaling) was used as a secondary antibody. Immuno-reactive bands were visualized by enhanced chemiluminescence (ECL; Amersham Biosciences Inc., Buckinghamshire, UK) and digitized using Labwork software (Analytik Jena AG, Jena, Thuringia, Germany).

### 2.8. Immunofluorescent (IF) Staining

The procedure and protocol have been described in our previous reports [34,35,36]. Re-hydrated paraffin sections were treated with 3% H_2_O_2_ for 30 min and incubated with Immuno-Block reagent (BioSB, Santa Barbara, CA, USA) for 30 min at room temperature. Sections were then incubated with primary antibodies against CD31 (1:100, Bio-Rad, Hercules, CA, USA), vascular endothelial growth factor (VEGF) (1:400, Abcam Inc., Cambridge, MA, USA), C-kit (1:100, Santa Cruz), GATA4 (1:500, Abcam) and SOX2 (1:100, Abcam). Three sections of heart specimens from each rat were analyzed. For quantification, three randomly selected HPFs (400× for IHC and IF studies) were analyzed in each section. The mean number of positively-stained cells per HPF for each animal was determined by summation of all numbers divided by 9.

### 2.9. Histological Quantification of Myocardial Size and Myocardial Fibrosis Area

The procedure and protocol have been described in detail in our previous reports [31,32,34]. Briefly, hematoxylin and eosin (H & E) and Masson’s trichrome staining were used to identify the cardiomyocyte size and fibrosis of LV myocardium, respectively. Three serial sections of LV myocardium in each animal were prepared at 4 µm thickness by Cryostat (Leica CM3050S, Leica Biosystems, Wetzlar, Germany). The integrated area (µm^2^) of fibrosis on each section was calculated using the Image Tool 3 (IT3) image analysis software (Image Tool for Windows, Version 3.0, University of Texas Health Science Center, San Antonio (UTHSCSA), TX, USA). Three randomly selected high-power fields (HPFs) (200×) were analyzed in each section. After determining the number of pixels in each fibrotic area per HPF, the number of pixels obtained from three HPFs were calculated. The procedure was repeated in two other sections for each animal. The mean pixel number per HPF for each animal was then determined by calculating all pixel numbers and divided by 9. The mean integrated area (µm^2^) of fibrosis in LV myocardium per HPF was obtained using a conversion factor of 19.24 (since 1 µm^2^ corresponds to 19.24 pixels).

### 2.10. Flow Cytometric Analysis for Assessment of Circulating Levels of Endothelial Progenitor Cells (EPCs) and Mesenchymal Stem Cells (MSCs) Based on Surface Markers

Blood sampling was performed for identification of peripheral blood-derived EPCs and MSC by flow cytometric analysis. The MSCs were stained with fluorescein isothiocyanate (FITC)- or phycoerythrin (PE)-conjugated antibodies against the following surface markers for flow cytometric analysis: CD29+/CD45−, CD73+/CD45−, CD90+/CD45− and CD105+/CD45− (BD Pharmingen, San Diego, CA, USA) in circulation. Additionally, the EPCs were immunostained for 30 min on ice with the following antibodies: PE-conjugated antibodies against CD133 (BD Pharmingen), Sca-1 (BD Pharmingen), and CD34 (BD Pharmingen); ffluorescein isothiocyanate (FITC)-conjugated antibodies against c-kit (BD Pharmingen); monoclonal antibodies against CD31 (abcam) VEGF (abcam), KDR (Thermo Fisher Scientific Inc., Waltham, MA, USA) and vascular endothelial cadherin (VE-Cad) (abacm). Cells labeled with non-fluorescence-conjugated antibodies were further incubated with Alexa Fluor 488-conjugated antibodies specifically against mouse or rabbit IgG (Thermo Fisher Scientific Inc., Waltham, MA, USA). Isotype-identical antibodies (IgG) served as controls. Flow cytometric analyses were performed by utilizing a fluorescence-activated cell sorter (Beckman Coulter FC500 flow cytometer, Beckman Coulter Inc., Brea, CA, USA).

### 2.11. Statistical Analysis

Quantitative data are expressed as mean ± SD. Statistical analysis was performed by one way ANOVA followed by Bonferroni multiple-comparison post hoc test. SAS statistical software for Windows version 8.2 (SAS Institute, Cary, NC, USA) was utilized. A probability value <0.05 was considered statistically significant.

## 3. Results

### 3.1. An Increase in Angiogenesis in Cellular and Protein Levels was Energy-Dependent Undergoing ECSW Therapy

To elucidate whether a stepwise increased ECSW dosage (i.e., increased energy dosage) would progressively enhance the angiogenesis ability in human umbilical vein endothelial cells (HUVECs), the Matrigel assay, Western blot and cell proliferative ability were employed in in vitro studies (Figure 1A–L). The results showed that the Matrigel assay of angiogenesis in HUVECs increased significantly stepwise as the dosage of ECSW energy progressively increased (Figure 1A–I). Additionally, the protein expressions of VEGF and bFGF, two soluble angiogenesis biomarkers, exhibited an identical pattern of Matrigel assay (Figure 1J,K). Furthermore, the protein expression of IGF-1 (Figure 1L), an indicator of stem cell growth factor, showed a similar pattern of Matrigel assay.

Additionally, the gene expression of ADMSC mitochondrial DNA also significantly progressively increased as the dosage of ECSW energy was upregulated (Figure 2F). Furthermore, the microscopic finding of IHC stain demonstrated that the number of ADMSC uptake of BrdU, an indicator of cell proliferation, also displayed a similar pattern of mitochondrial DNA among the four groups (Figure 2A–E). Our findings (i.e., in Figure 1 and Figure 2) suggested that with respect to the low-energy ECSW application, a notably positive correlation existed between the dosage of ECSW energy and angiogenesis capacity in HUVECs and proliferative capacity in ADMSCs.

### 3.2. ECSW Therapy Augmented Exogenous Mitochondria into ADMSCs

In vitro, IF microscopic findings (Figure 3(A1–D4)) showed that mitochondrial transfusion into ADMSCs upregulated the expression of exogenic mitochondria in these cells (Figure 3(A3–D3),E). Additionally, exogenic mitochondrial transfusion into ADMSCs was furthermore upregulated by ECSW therapy (Figure 3(A4–D4),G). These findings suggest that exogenous mitochondrial refreshment occurred in the recipient ADMSCs could be further upregulated by ECSW. The relative mitDNA from RT-PCR test for ADMSCs exhibited an identical pattern of IF microscopic findings among the groups (Figure 3H).

### 3.3. The ECSW Therapy Enhanced Circulating Levels of Endothelial Progenitor Cells (EPCs) and MSCs

To assess whether ECSW therapy would enhance the mobilization of EPCs from bone marrow into circulation, ECSW (1.5 mJ/mm^2^/200 shots) was applied to the femoral iliac bone areas of four additional rats. The result showed that, compared with the baseline, the circulating levels of c-Kit+/CD31+, Sca-1+/CD31+, vascular endothelial cadherin (Ve-Cad)+/CD34+ and KDR+/CD34+ cells (Figure 4A–D), four indicators of circulating EPCs and circulating levels of CD90+/CD45−, CD217+/CD45− and CD44+/CD45− cells (Figure 4E–G), three indicators of MSCs, were significantly increased by day 3 after ECSW therapy, suggesting that ECSW therapy, indeed, was able to upregulate the circulating levels of stem cells.

### 3.4. Cardiac Stem Cells (CSCs), Cardiac and Pluripotent Markers and Small Vessel Number Were Upregulated by Combined ECSW-Assisted Mitochondrial Delivery into ADMSCs + ECSW Therapy in LV Myocardium 7 Weeks after DCM Induction

To investigate whether the expressions of CSCs, cardiac and pluripotent markers were upregulated by ECSW-mito-ADMSCs therapy, staining of the LV specimen with immunofluorescence antibodies for detection of CSCs marker Sca-1 (Figure 5A–F), cardiac marker GATA4 (Figure 5G–L) and pluripotency marker SOX2 (Figure 6A–F) were performed. As we expected, these stem cell biomarkers were significantly progressively increased from groups 1 to 5. Additionally, the number of small vessels (i.e., diameters ≤ 25 μM) was highest in group 1, lowest in group 2 and significantly progressively increased from groups 3 to 5 (Figure 6G–L). These findings implicated two important messages, i.e., ECSW would arouse and combined ECSW-assisted mitochondrial delivery into ADMSCs + ECSW therapy would further arouse the expressions of CSCs as well as angiogenesis in DCM myocardium.

### 3.5. Combined ECSW-Assisted Mitochondrial Delivery into ADMSCs + ECSW Therapy Improved LVEF and Ameliorated the Heart Failure/Pressure Overload Biomarkers in LV Myocardium 7 Weeks after DCM Induction

To identify whether this innovative therapy would improve the LVEF and attenuate the heart failure biomarkers in LV myocardium, cardiac echo and Western blot analysis were performed. The result demonstrated that the LVEF did not differ by day 0 prior to DCM induction (Figure 7A). However, by 7 weeks after DCM induction, the LVEF was highest in group 1, lowest in group 2 and significantly higher in group 5 than in groups 3 and 4, but this parameter did not differ between groups 3 and 4 (Figure 7B).

Further, the Western blot result showed that the protein expression of BNP (Figure 7C), an indicator of pressure overload/heart failure biomarker, and the protein expression of ß-MHC (Figure 7D), an indicator of cardiomyocyte hypertrophy, were lowest in group 1 and significantly progressively reduced from groups 2 to 5. On the other hand, the protein expression of α-MHC (Figure 7E), a reversed cardiomyocyte hypertrophic biomarker, exhibited an opposite pattern of ß-MHC among the groups.

### 3.6. Combined ECSW-Assisted Mitochondrial Delivery into ADMSCs + ECSW Therapy Upregulated Angiogenesis in LV Myocardium of 7 Weeks after DCM Induction

To elucidate whether combined ECSW-assisted mitochondrial delivery into ADMSCs + ECSW therapy would augment the angiogenesis, IF stain and Western blot were utilized. IF microscopic finding showed that the cellular expression of CD31, an indicator of the integrity of endothelial cell function, was highest in group 1 and significantly progressively increased from groups 2 to 5 (Figure 8A–F). On the other hand, the cellular expression of VEGF, an indicator of angiogenesis, was progressively increased from groups 1 to 5, implicating that an intrinsic response to DCM injury was upregulated by ECSW-mito-ADMSCs therapy (Figure 8G–L). Additionally, the protein expression of CD31 exhibited an identical pattern of cellular expression of CD31 (Figure 8M), whereas the protein expression of VEGF (Figure 8N) also displayed an identical pattern of cellular expression of VEGF among the five groups.

### 3.7. Combined ECSW-Assisted Mitochondrial Delivery into ADMSCs + ECSW Therapy Downregulated Apoptosis, Fibrosis and Oxidative Stress and Preserved Mitochondrial Integrity in LV Myocardium 7 Weeks after DCM Induction

Western blot assessment was utilized for determination of apoptosis, fibrosis and oxidative stress. The results showed that the protein expressions of cleaved caspase 3 (Figure 9A), cleaved PARP (Figure 9B) and mitochondrial Bax (Figure 9C), three indices of apoptosis, and protein expressions of Smad3 (Figure 9D) and TGF-ß (Figure 9E), two indices of fibrosis, were lowest in group 1, highest in group 2 and significantly progressively decreased from group 3 to 5.

The protein expression of cytosolic cytochrome C (Figure 9F), an indicator of mitochondrial damage, and protein expressions of NOX-1 (Figure 9G) and NOX-2 (Figure 9H), two indicators of oxidative stress, displayed an identical pattern of apoptosis, whereas, the protein expression of mitochondrial cytochrome C (Figure 9I), an indicator of mitochondrial integrity, exhibited an opposite pattern of apoptosis among the groups.

### 3.8. Cardiomyocyte Size and Cellular Expression of Fibrosis among the Five Groups by Day 49 after DCM Induction

To verify the changes of cellular expressions of cardiomyocytes and fibrosis in LV myocardium, the H.E. stain and IHC stain were utilized in the present study. As we expected, the cardiomyocyte size, an index of hypertrophy, was lowest in group 1, highest in group 2, and then significantly progressively decreased from groups 3 to 5 (Figure 10A–F). Consistently, the fibrotic area exhibited an identical pattern of cardiomyocyte size among the groups (Figure 10G–L).

## 4. Discussion

This study, which investigated the therapeutic impact of ECSW-mito-ADMSCs on protecting the cardiac structural and functional integrities against DCM damage, yielded several striking preclinical implications. First, the study demonstrated that, compared with the SC group, the LVEF was remarkably reduced, suggesting that our DCM model in rodent was successfully created for individual study. Second, therapy with ECSW-assisted mito delivery into ADMSCs was better than ECSW only for protecting the heart against DCM damage. Third, a combined therapy by ECSW-assisted mito delivery into ADMSCs implantation to LV myocardium followed by additional ECSW to the DCM heart was superior to other strategic modalities for preserving the heart architecture and functional integrity.

Abundant data have demonstrated that ECSW therapy enhanced angiogenesis, restored blood flow and improved ischemic-related organ dysfunction [31,32,33,37]. An essential finding in the present study was that ECSW therapy enhanced cellular and protein levels of angiogenesis and improved functional integrity of endothelial cells. Additionally, there was a significantly positive correlation between increased dosage of ECSW energy and increased capacity of angiogenesis in both in vitro and in vivo studies. Furthermore, ECSW therapy also enhanced the mobilization of EPCs from bone marrow to circulation. On the other hand, we also found that ECSW-assisted mito delivery into ADMSCs therapy was better than ECSW alone for enhancement of angiogenesis in both in vitro and in vivo situations. Accordingly, our findings extended the findings of previous studies [31,32,33,37].

A study has previously revealed that ECSW therapy upregulated the number of NeuN+ cells in animals with chronic cerebral hypoperfusion syndrome [38]. Additionally, other experimental studies have also shown that ECSW treatment augmented cell/MSCs proliferation and migratory activity [39,40]. Another essential finding in the present study was that ECSW therapy enhanced ADMSCs differentiation/proliferation (i.e., in vitro study). Additionally, this therapy was further identified to have the capacity of upregulating the CSCs differentiation (i.e., in vivo study). In this way, our results corroborated with the findings of previous studies [38,39]. Intriguingly, our study further demonstrated that ECSW-assisted mito delivery into ADMSCs was better, and ECSW-assisted mito delivery into ADMSCs implantation to LV myocardium followed by additional ECSW therapy was even better than ECSW therapy only on upregulating CSCs differentiation (i.e., in vivo study). In this way, our findings extended the findings of previous studies [38,39].

Compared to the SC group, the LVEF was markedly impaired in DCM animals. Additionally, cellular level of fibrosis, cardiomyocytes size as well as the protein levels of the heart failure/pressure overload, fibrosis, apoptotic and oxidative stress biomarkers were remarkably increased in DCM animals compared to the SC group. These findings, in addition to supporting our success in creating the DCM model, could explain why the LVEF was more impaired in DCM animals than in their SC counterparts. The most important finding in the present study was that those aforementioned molecular-cellular perturbations were substantially attenuated by ECSW (i.e., DCM with first therapeutic modality), further substantially attenuated by ECSW-assisted mito delivery into ADMSCs (DCM with second therapeutic modality) and furthermore substantially attenuated by ECSW-assisted mito delivery into ADMSCs implantation to LV myocardium followed by additional ECSW (DCM with third modality) therapies. These findings could explain why, by day 49 (i.e., the end of 7 weeks) after DCM induction, the LVEF was progressively improved from first to third type therapy.

One thing worthy of pointing out is that our previous studies have proved that exogenous mitochondria could be transferred into recipient cells in in vitro condition [35,36]. Additionally, ECSW therapy enhanced exogenous mitochondria into the cells not only in in vitro condition [35,36], but also in in vivo situation [35]. In the present study, we also found that ECSW-enhanced exogenous mitochondria into ADMSCs demonstrates a superior effectiveness on preserving the cardiac function in setting of DCM, highlighting that this may pose a therapeutic potential in our future clinical practice for DCM patients, especially those who have poor LV function and are resisting to conventional therapy.

We remain uncertain why ECSW therapy would further enhance the exogenous mitochondria into ADMSCs. However, results of previous studies may shed light on this question. First, our previous study [39] has shown that the ECSW therapy activated plasma membrane/intracellular compartment and its ultrastructure, and upregulated intracellular mechanotransduction signaling axis [40] that further upregulated focal adhesion kinase/mTOR signaling for intracellular transports, and consequently stimulated the MSCs proliferation. Second, another of our studies has demonstrated that ECSW therapy was able to augment cellular endocytosis [39]. All this led us to speculate that “intracellular mechanotransduction” and “endocytosis” could, at least in part, explain why application of ECSW to the recipient cells could enhance cells uptake of exogenous mitochondria.

## 5. Conclusions

This study demonstrated that therapy with ECSW-assisted mito delivery into ADMSCs was better than ECSW alone for protecting the heart against DCM damage. It also showed that combined therapy by ECSW-assisted mito delivery into ADMSCs implantation to LV myocardium followed by additional ECSW to the DCM heart was superior to other therapeutic strategies for further preserving the cardiac structural and functional integrity.

### 5.1. Study Limitation

This study has limitations. First, the study period was 49 days, implicating that it was relatively short when DCM was utilized as an experimental model of a specific disease entity. Accordingly, the long-term impact of ECSW-mito-ADMSCs therapy on preserving the heart function in DCM setting remains uncertain. Second, this study did not yet present a direct piece of evidence and underlying mechanism to explain how the ECSW therapy could enhance the exogenous mitochondria transferred into the recipient cells. Third, this study did not perform experiments to assess the intracellular calcium handling, calcium wave or tension for understanding the Ca^2+^ ionic currents or voltage across the cell membrane undergoing the ECSW therapy. Finally, although extensive work had been accomplished in the present study, the underlying mechanism for ECSW-facilitated mito into ADMSCs therapy on improving the heart function in DCM setting might not be completely elucidated. Accordingly, the preliminary proposals regarding the underlying mechanism of these therapies on improving the heart function, based on our findings, were schematically presented in Figure 11.

### 5.2. In Conclusion

Results of the present study demonstrated that ECSW-mito-ADMSCs therapy effectively improved heart function in DCM rat. The results of this study, which well documented the underlying mechanism on improving heart function, were schematically proposed in Figure 11.

## Figures and Tables

**Figure 1 biomedicines-09-01362-f001:**
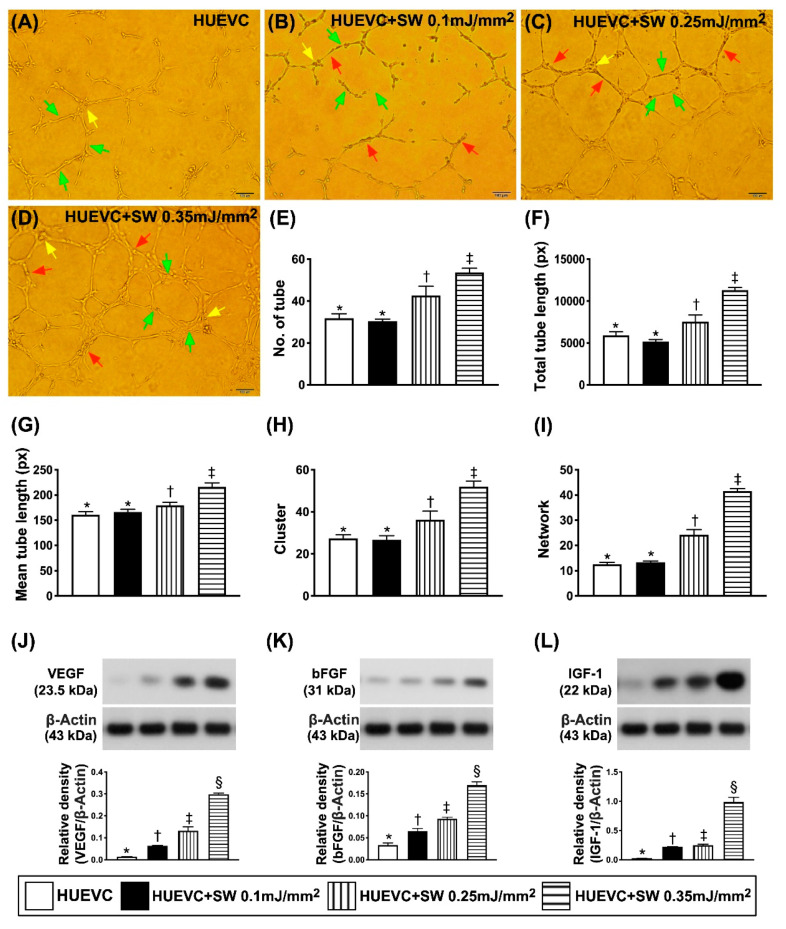
A positive correlation between increased ECSW energy and increased angiogenesis from both HUVECs and ADMSCs. (**A**–**D**) Illustrating (100×) the Matrigel assay for identification of a stepwise increase in ECSW dosages and upregulated capacity of angiogenesis in human umbilical vein endothelial cells (HUVECs). The parameters of angiogenesis, including: (1) tubular formation (red arrows), (2) cluster formation (yellow arrows) and (3) network formation (green color). Scale bar in right lower corner represents 100 µm. (**E**) Analytical result of number of tubules, * vs. other groups with different symbols (†, ‡), *p* < 0.001. (**F**) Analytical result of total tubular length, * vs. other groups with different symbols (†, ‡), *p* < 0.001. (**G**) Analytical result of mean tubular length, * vs. other groups with different symbols (†, ‡), *p* < 0.001. (**H**) Analytical result of cluster formation * vs. other groups with different symbols (†, ‡), *p* < 0.001. (**I**) Analytical result of network formation, * vs. other groups with different symbols (†, ‡), *p* < 0.001. (**J**) Protein expression of vascular endothelial growth factor (VEGF) in ADMSCs, * vs. other groups with different symbols (†, ‡, §), *p* < 0.001. (**K**) Protein expression of basic fibroblast growth factor (bFGF) in ADMSCs, * vs. other groups with different symbols (†, ‡, §), *p* < 0.001. (**L**) Protein expression of insulin-like growth factor (IGF)-1 in ADMSCs, * vs. other groups with different symbols (†, ‡, §), *p* < 0.001. All statistical analyses were performed by one-way ANOVA, followed by Bonferroni multiple comparison post hoc test (*n* = 6 for each group). ADMSCs = adipose-derived mesenchymal stem cells; SW = shock wave.

**Figure 2 biomedicines-09-01362-f002:**
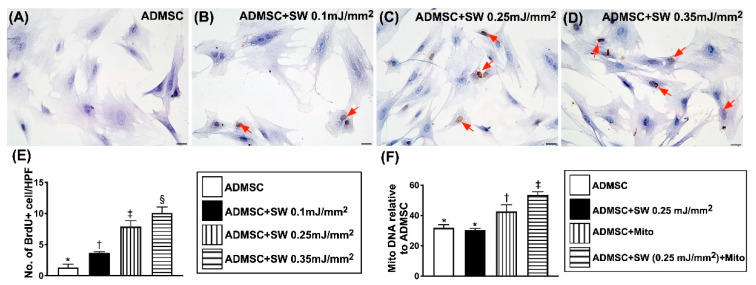
ECSW therapy enhanced ADMSCs proliferation and ECSW-assisted exogenous mitochondria transferred into recipient ADMSCs. (**A**–**D**) Illustrating microscopic finding (400×) of immunohistochemical (IHC) stain for identifying a progressively increased appearance of BrdU uptake (gray color) (red arrows) after a stepwise increase in dosages of ECSW energy. Scale bar in right lower corner represents 20 µm. (**E**) Number of BrdU expression per higher-power field (HPF), * vs. other groups with different symbols (†, ‡, §), *p* < 0.0001. (**F**) Gene expression of ADMSC mitochondrial DNA among the groups, * vs. other groups with different symbols (†, ‡), *p* < 0.0001. All statistical analyses were performed by one-way ANOVA, followed by Bonferroni multiple comparison post hoc test (*n* = 6 for each group). ADMSCs = adipose-derived mesenchymal stem cells; SW = shock wave.

**Figure 3 biomedicines-09-01362-f003:**
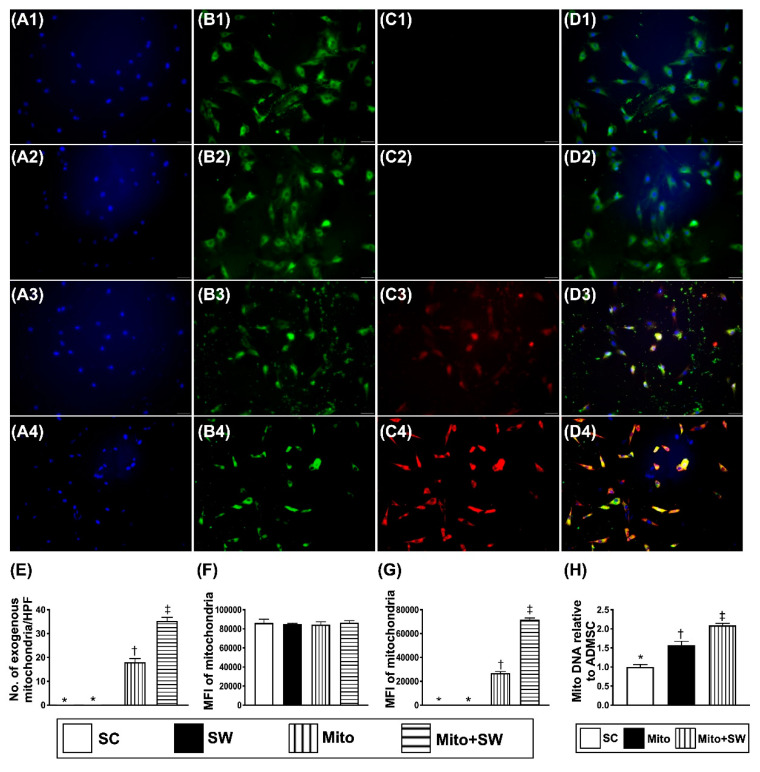
ECSW therapy augmented exogenous mitochondria into the rat ADMSCs. (**A1**–**A4**) Showing the DAPI stain (400×) for verifying ADMSCs in four groups (i.e., SC, SW, mito and mito + SW (0.2 mJ/mm^2^ for 100 shots)). (**B1**–**B4**) Demonstrating MitoTracker stain (400×) for identifying the endogenous mitochondria (green color) among three groups. (**C1**–**C4**) Illustrating the MitoTracker stain (400×) for identification of exogenous mitochondria transferred into recipient ADMSCs (red color). (**D1**–**D4**) Indicating the merged picture of B and C. Pink-yellow color represented the endogenous and exogenous mitochondria colocalized together. Plentiful mitochondria were found in mito + SW group. The scale bar in right lower corner of A1–D4 represents 20 µm. (**E**) Analytic result of number of exogenous mitochondria in the ADMSCs, * vs. other groups with different symbols (†, ‡), *p* < 0.001. (**F**) Analytical result of fluorescent intensity of endogenous mitochondria (indicated green fluorescent protein (GFP) of (**B1**–**B4**)), *p* > 0.5. (**G**) Analytical result of fluorescent intensity of exogenous mitochondria (indicated green fluorescent protein (GFP) of (**C1**–**C4**)), * vs. other groups with different symbols (†, ‡), *p* < 0.001. (**H**) Analytical result of RT-PCR of relative mitDNA, * vs. other groups with different symbols (†, ‡), *p* < 0.001. Scale bar in right lower corner represents 20 µm. All statistical analyses were performed by one-way ANOVA, followed by Bonferroni multiple comparison post hoc test (*n* = 6 for each group). Symbols (*, †, ‡) indicate significance (at 0.05 level). HPF = high-power field; MIF = mean fluorescent intensity; ADMSCs = adipose-derived mesenchymal stem cells; SC = sham control; mito = mitochondria; SW = shock wave.

**Figure 4 biomedicines-09-01362-f004:**
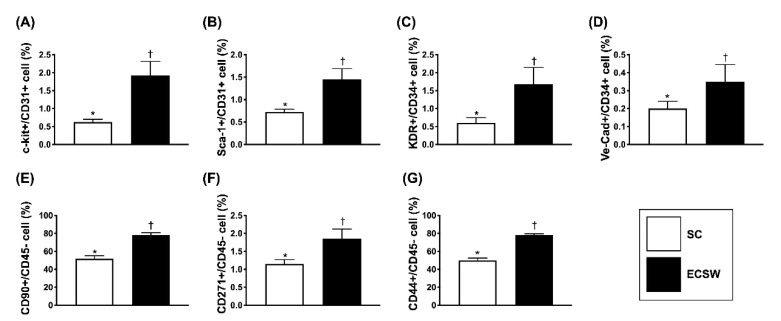
Circulating levels of endothelial progenitor cells (EPCs) and MSCs by 72h after ECSW therapy. (**A**) Analytical result of number of circulating c-kit +/CD133+ cells, * vs. †, *p* < 0.01. (**B**) Analytical result of number of circulating Sca-1 +/CD31 + cells, * vs. †, *p* < 0.01. (**C**) Analytical result of number of circulating KDR+/CD34+ cells, * vs. †, *p* < 0.01. (**D**) Analytical result of number of circulating Ve-Cad+/CD34+ cells, * vs. †, *p* < 0.01. (**E**) Analytical result of number of circulating CD90+/CD45− cells, * vs. †, *p* < 0.01. (**F**) Analytical result of number of circulating CD271+/CD45− cells, * vs. †, *p* < 0.01. (**G**) Analytical result of circulating CD44+/CD45− cells, * vs. †, *p* < 0.01. Ve-Cad = vascular endothelial cadherin; SC = sham control; mito = mitochondria; ECSW = extracorporeal shock wave.

**Figure 5 biomedicines-09-01362-f005:**
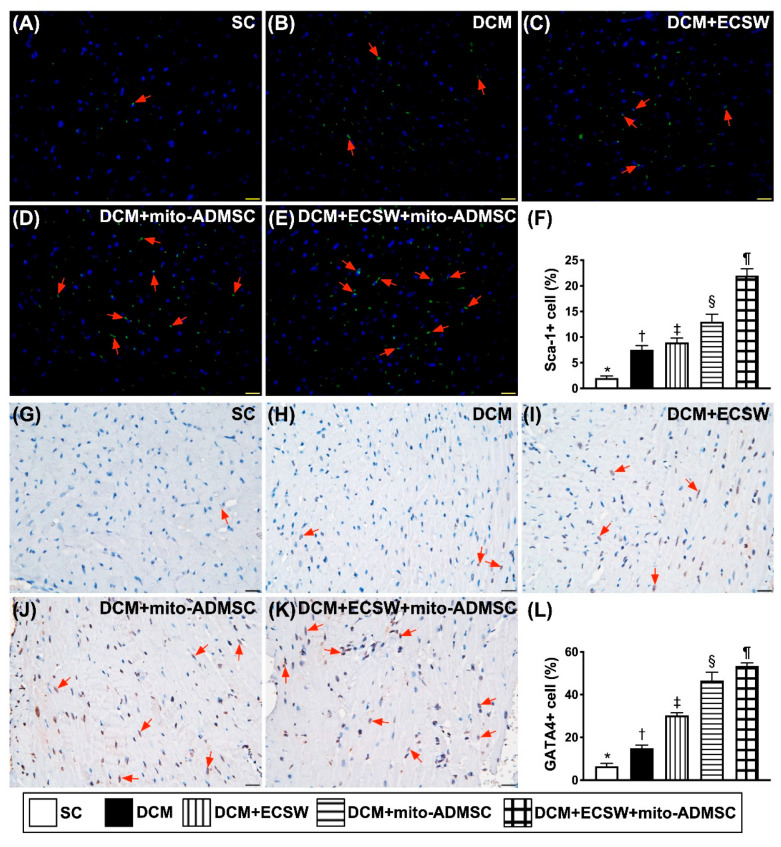
Cardiac stem cells and cardiac marker were upregulated by combined ECSW-assisted mitochondrial delivery into ADMSCs + ECSW therapy in LV myocardium 7 weeks after DCM induction. (**A**–**E**) Illustrating the microscopic finding (400×) for identification of cellular expression of Sca-1 (green color) (red arrows). (**F**) Analytical result of number of Sca-1+ cells, * vs. other groups with different symbols (†, ‡, §, ¶), *p* < 0.0001. (**G**–**K**) Illustrating the microscopic finding (400×) for identification of cellular expression of GATA4 (gray color) (red arrows). (**L**) Analytical result of number of GATA4 + cells, * vs. other groups with different symbols (†, ‡, §, ¶), *p* < 0.0001. Scale bar in right lower corner represents 20 µm. All statistical analyses were performed by one-way ANOVA, followed by Bonferroni multiple comparison post hoc test (*n* = 6 for each group). Symbols (*, †, ‡, ¶) indicate significance (at 0.05 level). ECSW = extracorporeal shock wave; SC = sham control; DCM = dilated cardiomyopathy; mito = mitochondria.

**Figure 6 biomedicines-09-01362-f006:**
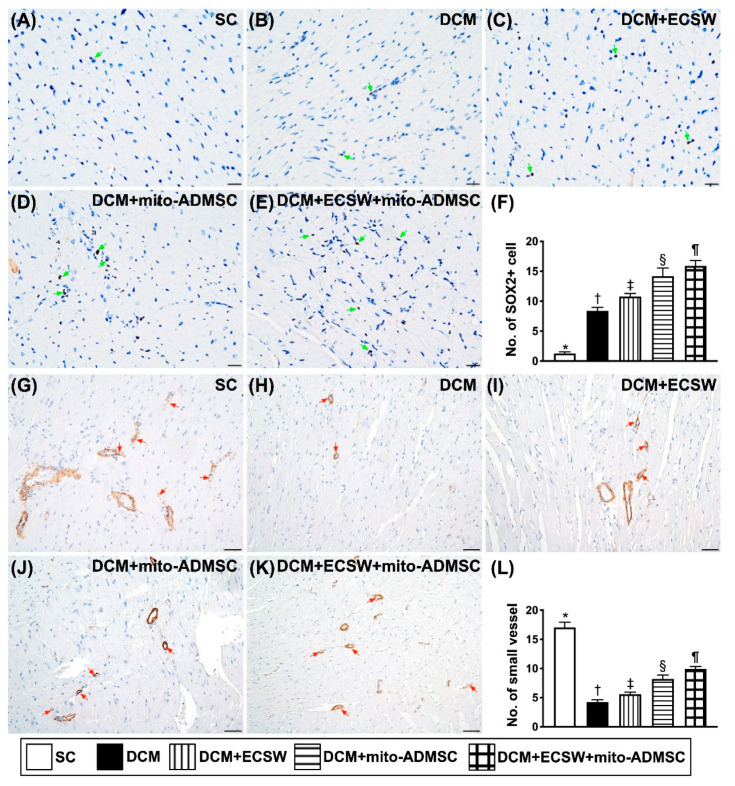
Pluripotency markers were upregulated by combined ECSW-assisted mitochondrial delivery into ADMSCs + ECSW therapy in LV myocardium 7 weeks after DCM induction. (**A**–**E**) Illustrating the microscopic finding (400×) for identification of cellular expression of SOX2 (gray color) (green arrows). (**F**) Analytical result of number of SOX2+ cells, * vs. other groups with different symbols (†, ‡, §, ¶), *p* < 0.0001. (**G**–**K**) Illustrating the microscopic finding (400×) of α-smooth muscle actin (α-SMC) positive stain for identification of small vessel (i.e., diameter ≤ 25 μM) density (gray color) (red color). (**L**) Analytical result of number of small vessels, * vs. other groups with different symbols (†, ‡, §, ¶), *p* < 0.0001. Scale bar in right lower corner represents 20 µm for SOX2 and 50 µm for α-SMC positive staining, respectively. All statistical analyses were performed by one-way ANOVA, followed by Bonferroni multiple comparison post hoc test (*n* = 6 for each group). Symbols (*, †, ‡, ¶) indicate significance (at 0.05 level). SC = sham control; DCM = dilated cardiomyopathy; ECSW = extracorporeal shock wave; mito = mitochondria.

**Figure 7 biomedicines-09-01362-f007:**
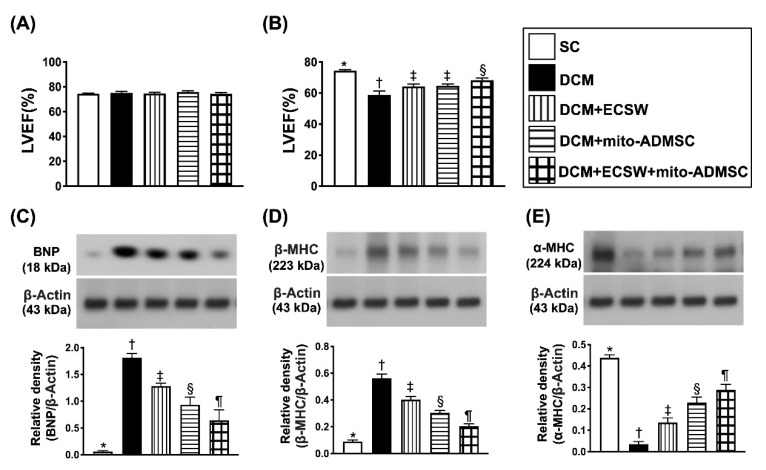
Heart function and biomarkers of heart failure and pressure overload in LV myocardium 7 weeks after DCM induction. (**A**) Left ventricular ejection fraction (LVEF) by day 0, *p* > 0.5. (**B**) LVEF by day 49, * vs. other groups with different symbols (†, ‡, §), *p* < 0.0001. (**C**) Protein expression of brain natriuretic peptide (BNP), * vs. other groups with different symbols (†, ‡, §, ¶), *p* < 0.0001. (**D**) Protein expression of beta myelin heavy chain (ß-MHC), * vs. other groups with different symbols (†, ‡, §, ¶), *p* < 0.0001. (**E**) Protein expression of α-MHC, * vs. other groups with different symbols (†, ‡, §, ¶), *p* < 0.0001. All statistical analyses were performed by one-way ANOVA, followed by Bonferroni multiple comparison post hoc test (*n* = 6 for each group). Symbols (*, †, ‡, ¶) indicate significance (at 0.05 level). SC = sham control; DCM = dilated cardiomyopathy; ECSW = extracorporeal shock wave; mito = mitochondria.

**Figure 8 biomedicines-09-01362-f008:**
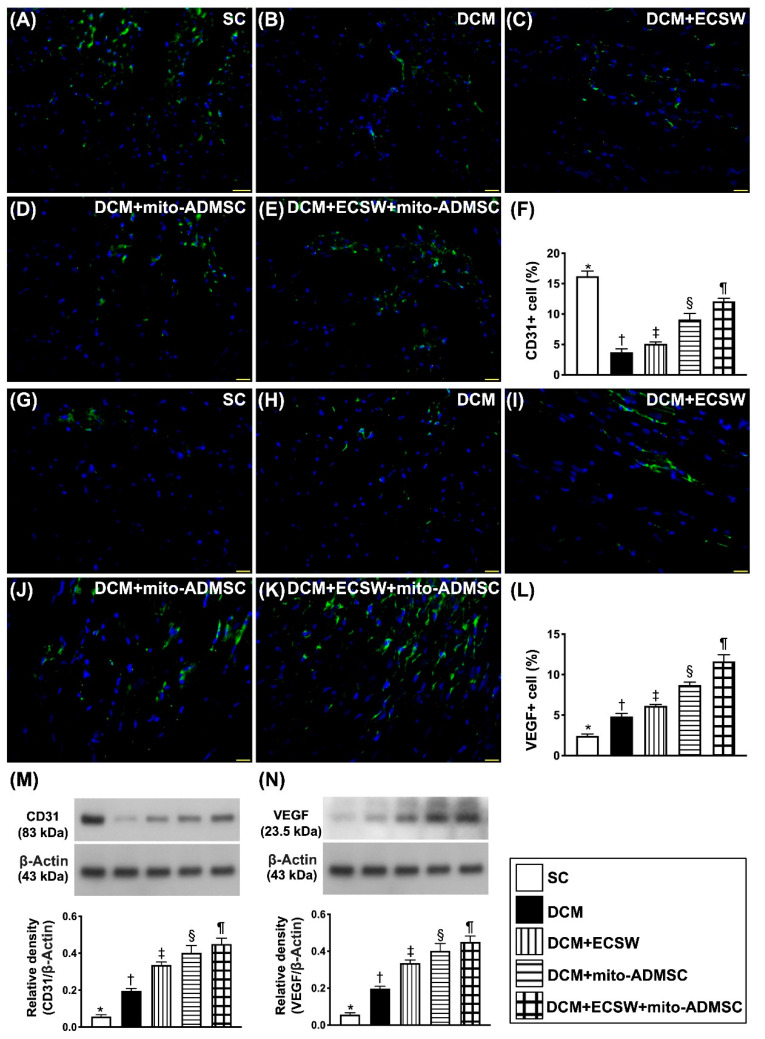
Angiogenesis in LV myocardium 7 weeks after DCM induction. (**A**–**E**) Illustrating the immunofluorescent (IF) microscopic finding (400×) for identification of cellular expression of CD31 (green color). (**F**) Analytical result of number of CD31+ cells, * vs. other groups with different symbols (†, ‡, §, ¶), *p* < 0.0001. (**G**–**K**) Illustrating the IF microscopic finding (400×) for identification of cellular expression of vascular endothelial growth factor (VEGF) (green color). (**L**) Analytical result of number of VEGF+ cells, * vs. other groups with different symbols (†, ‡, §, ¶), *p* < 0.0001. Scale bar in right lower corner represents 50 µm. (**M**) Protein expression of CD31, * vs. other groups with different symbols (†, ‡, §, ¶), *p* < 0.0001. (**N**) Protein expression of VEGF, * vs. other groups with different symbols (†, ‡, §, ¶), *p* < 0.0001. All statistical analyses were performed by one-way ANOVA, followed by Bonferroni multiple comparison post hoc test (*n* = 6 for each group). Symbols (*, †, ‡, ¶) indicate significance (at 0.05 level). SC = sham control; DCM = dilated cardiomyopathy; ECSW = extracorporeal shock wave; mito = mitochondria.

**Figure 9 biomedicines-09-01362-f009:**
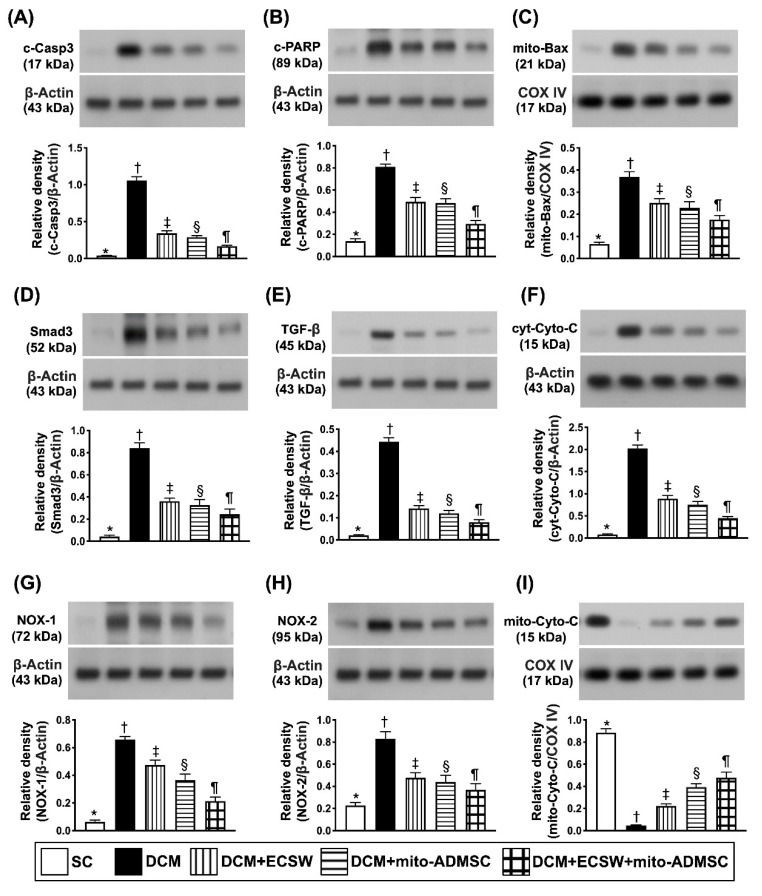
Apoptosis, fibrosis, oxidative stress and mitochondrial damage marker in LV myocardium 7 weeks after DCM induction. (**A**) Protein expression of cleaved caspase 3 (c-Casp3), * vs. other groups with different symbols (†, ‡, §, ¶), *p* < 0.0001. (**B**) Protein expression of c-PARP, * vs. other groups with different symbols (†, ‡, §, ¶), *p* < 0.0001. (**C**) Protein expression of mitochondrial (mito)-Bax, * vs. other groups with different symbols (†, ‡, §, ¶), *p* < 0.0001. (**D**) Protein expression of Smad3, * vs. other groups with different symbols (†, ‡, §, ¶), *p* < 0.0001. (**E**) Protein expression of transforming growth factor (TGF)-ß, * vs. other groups with different symbols (†, ‡, §, ¶), *p* < 0.0001. (**F**) Protein expression of cytosolic cytochrome C (cyt-Cyto-C), * vs. other groups with different symbols (†, ‡, §, ¶), *p* < 0.0001. (**G**) Protein expression of NOX-1, * vs. other groups with different symbols (†, ‡, §, ¶), *p* < 0.0001. (**H**) Protein expression of NOX-2, * vs. other groups with different symbols (†, ‡, §, ¶), *p* < 0.0001. (**I**) Protein expression of mitochondrial cytochrome C (mito-Cyto-C), * vs. other groups with different symbols (†, ‡, §, ¶), *p* < 0.0001. All statistical analyses were performed by one-way ANOVA, followed by Bonferroni multiple comparison post hoc test (*n* = 6 for each group). Symbols (*, †, ‡, ¶) indicate significance (at 0.05 level). SC = sham control; DCM = dilated cardiomyopathy; ECSW = extracorporeal shock wave; mito = mitochondria.

**Figure 10 biomedicines-09-01362-f010:**
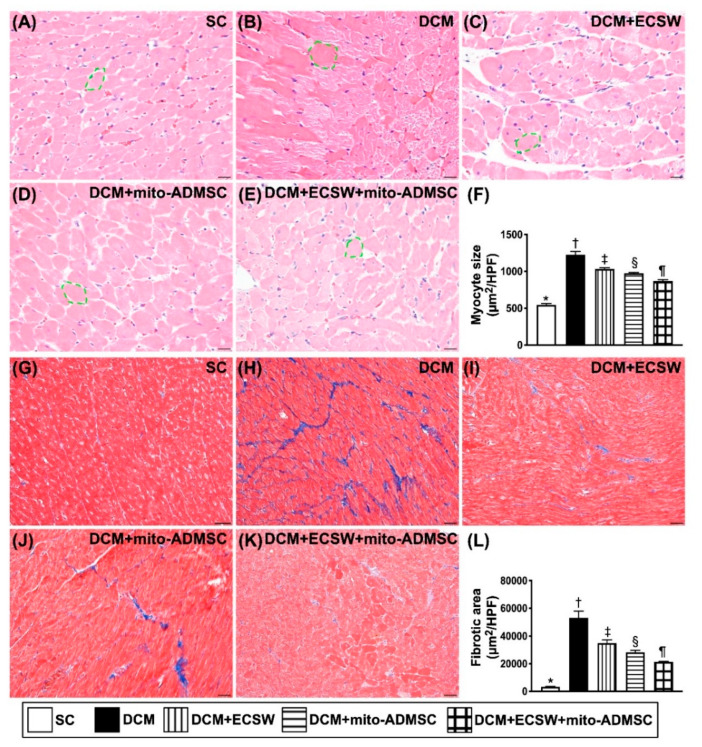
Cardiomyocyte size and cellular expression of fibrosis by day 49 after DCM induction. (**A**–**E**) Illustrating the microscopic finding (200×) of H.E. stain for identification of cardiomyocyte size (yellow dotted line). (**F**) Analytical result of cardiomyocyte size, * vs. other groups with different symbols (†, ‡, §, ¶), *p* < 0.0001. (**G**–**K**) Illustrating the immunohistochemical stain (200×) for identification of fibrosis in LV myocardium (blue color). (**L**) Analytical result of fibrotic area, * vs. other groups with different symbols (†, ‡, §, ¶), *p* < 0.0001. Scale bar in right lower corner represents 50 µm. All statistical analyses were performed by one-way ANOVA, followed by Bonferroni multiple comparison post hoc test (*n* = 6 for each group). Symbols (*, †, ‡, ¶) indicate significance (at 0.05 level). SC = sham control; DCM = dilated cardiomyopathy; ECSW = extracorporeal shock wave; mito = mitochondria.

**Figure 11 biomedicines-09-01362-f011:**
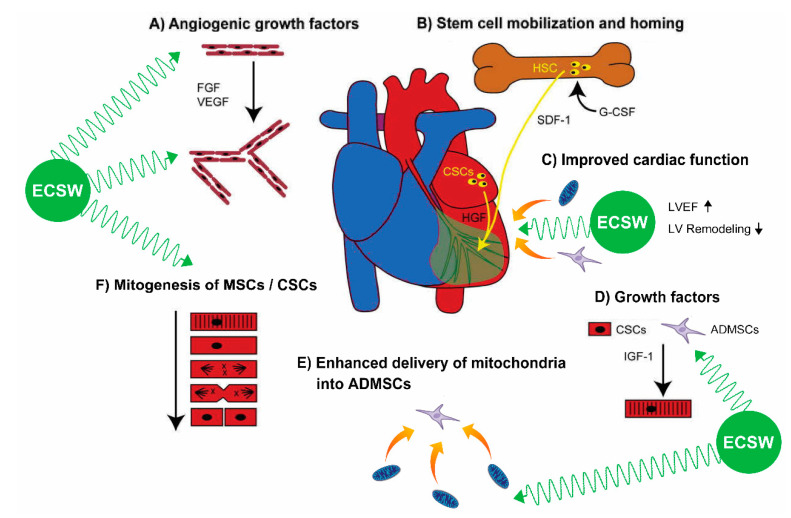
Summary demonstrating that ECSW-facilitated exogenous mitochondria into ADMSCs therapy ameliorated the LV function mainly through enhancing angiogenesis, expression of cardiac stem cells/growth factors in LV myocardium, mobilization of bone marrow-derived endothelial progenitor cells into circulation, MSC differentiation and soluble angiogenesis factors. ADMSCs = adipose-derived mesenchymal stem cells; ECSW = extracorporeal shock wave; DCM = dilated cardiomyopathy; LVEF = left ventricular ejection fraction; CSC = cardiac stem cell; VEGF = vascular endothelial growth factor; TGF-ß = transforming growth factor beta.

## Data Availability

The datasets of present study can be available from the corresponding author upon request.

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
