# Peer review of "Extracorporeal Shock Wave Enhanced Exogenous Mitochondria into Adipose-Derived Mesenchymal Stem Cells and Further Preserved Heart Function in Rat Dilated Cardiomyopathy"

_biomedicines, 2021, doi:10.3390/biomedicines9101362_

Round 1

Reviewer 1 Report

In this experimental study, Dr. Sung and colleagues investigated the effects of extracorporeal shockwave (ECSW) and exogenous mitochondria on ADMSCs in improving dilated cardiomyopathy. Overall, the study was done comprehensively and the manuscript was nicely written. The study looks promising to pursue further as well. However, some parts require clarifications and corrections, and I also have some suggestions to be incorporated:

  • Lines 69-74: LV contractile dysfunction is most likely caused by derangement or abnormality of cardiomyocyte calcium handling (PMID: 32188566). This important statement needs to be specified in the paragraph and could be extensively discussed. Everything else mentioned by the authors (from inflammation to scar) could indeed happen but how they affect contractile machinery is missing and was not explained. Please discuss extensively how those factors could influence contractile function of working cardiomyocytes.
  • Line 233: "The results showed that the Matrigel assay of angiogenesis in HUVECs was significantly stepwise increased as the dosage of ECSW energy progressively increased", Is there any maximum / saturating dose of ECSW energy with regards to angiogenesis? It would be insightful to add more dosage until it reaches the saturating dose. Please perform the experiments if possible.
  • I noticed that the doses of shockwave used in Figure 1, 2, 3 (0.x mJ) were significantly lower as compared to Figure 4 onwards (1.5 mJ). What was the reason for this? Please elaborate because it could be that such dose difference would affect the interpretability of the findings.
  • Line 354 and Figure 7B: when I see the LVEF reduction, it seems to me that although the number is lower than control, it is still within the normal range (60%). Do the authors believe that this model could sufficiently represent DCM (as claimed by the authors in line 443 and 475), which has a contractile dysfunction and most likely result in LVEF below normal range? This definitely require a justification and I am also wondering why didn't the authors impair the LVEF further with doxorubicin? I am sure it is doable. 
  • Also, the increase with both shockwave and exogenous mitochondria showed roughly 5% increase of LVEF. Do the authors think that 5% of LVEF improvement would be clinically relevant? Would there be a significant improvement on the NYHA class?
  • In the manuscript, although the authors have demonstrated extensively the effects of shockwave and exogenous mito-ADMSCs on structural remodeling, it lacks of functional investigations showing that those interventions also improve cardiomyocyte physiology, not only structures. The only functional assessment was done by measuring LVEF. However, it is not adequate since LVEF is not a cellular property. The authors could perform experiments to assess the intracellular calcium handling or calcium wave or tension, etc.
  • I am not sure if ADMSC is a common abbreviation, which doesn't require any description in its first use. Therefore, adding it in the title without describing what it is would not be appropriate. Moreover, in the abstract the description of ADMSC (what it stands for) is not available. Please adapt the title so that it is well explained while also maintaining the length of the title within a reasonable limit.
  • Section 0 needs to be removed. 
  • In general, the introduction needs to be more concise. Some words are not needed and can be removed to improve clarity of the content. 
    • "It is well recognized that the dilated cardiomyopathy (DCM), a non-ischemia related heart muscle disease with architectural..."
    • "The DCM is a clinical diagnosis characterized by left ventricular or biventricular dilation and impaired contraction in the absence of coronary artery disease, hypertension, valvular disease or congenital..."
    • Line 53: Please check if "anomaly" is the most appropriate word. In my opinion, "abnormality" is more appropriate.
    • "Genetic inheritance arises in 30-48% of patients, and inflammatory disorders such as myocarditis or toxic effects from medications, alcohol, or illicit drugs also could result in dilated cardiomyopathy"
    • Line 69: Please choose one between "dilation" and "dilatation". Check the whole manuscript and make it consistent.
    • "our previous studies have revealed that combined ADMSCs and ECSW therapy was superior to either one for improving acute ischemia-related LV dysfunction"
    • Line 88: "ADMASCs"
    • Line 89: not sure if this is grammatically correct "mitochondrially dysfunctional cardiomyocytes" Please rephrase.
  • An important remark: please make sure to remove "Figure X" from all (sub)headings and when linking the figure to the text, please specify which part of the figures we should look into. Don't say "Fig 1" only, but please use "Fig 1A" or "Fig 1B", etc.
  • Heading of section 3.2: "ECSW therapy augmented exogenous mitochondria into ADMSCs" augmented to do what? internalization? diffusion? Please clarify.
  • Figure 5 needs arrows as seen in other figures to point out the Sca-1 (green) and GATA4 (grey). They are not clearly visible and readers might have difficulty confirming whether the numbers are indeed consistent with the data shown in the bar charts.
  • Line 340: "Illustrating the microscopic finding (400x) of a-smooth muscle actin (a-SMC) positive stain for identification of small vessel (i.e., diameter = 25µM) density (gray color) (red color)." I am curious, how did the authors quantify the small vessels? As far as I know, it is highly dependent on the cutting/slicing angle and therefore 2 vessels in the preparation could be from one single vessel which is cut twice in a different location?
  • Not sure why there are two conclusions in this manuscript. Please merge them or remove one. 
  • Figure 11 needs to be linked with the text. Otherwise, this could be a nice graphical abstract. The figure also require a caption explaining what we see in the figure.

Author Response

Response to Reviewer's Comments (Reviewer #1)

Dear Reviewer:

Your constructive criticism is greatly appreciated. We have made the following responses to comply with your honorable suggestions (Note: The revised parts of the manuscript in response to Reviewer’s comments have been marked in red color):

Response to comments

Comment 1: Lines 69-74: LV contractile dysfunction is most likely caused by derangement or abnormality of cardiomyocyte calcium handling (PMID: 32188566). This important statement needs to be specified in the paragraph and could be extensively discussed. Everything else mentioned by the authors (from inflammation to scar) could indeed happen but how they affect contractile machinery is missing and was not explained. Please discuss extensively how those factors could influence contractile function of working cardiomyocytes.

Response 1: Yes, according to your recommendation, we have discussed extensively for how those factors could influence contractile function of working cardiomyocytes.

Comment 2: Line 233: "The results showed that the Matrigel assay of angiogenesis in HUVECs was significantly stepwise increased as the dosage of ECSW energy progressively increased", Is there any maximum / saturating dose of ECSW energy with regards to angiogenesis? It would be insightful to add more dosage until it reaches the saturating dose. Please perform the experiments if possible.

Response 2: Dear reviewer, our recent study had demonstrated that although ECSW energy of 0.35 mJ/mm2 upregulated the angiogenesis in HUVECs, the cell proliferation (i.e., by MTT assay) rat started to be suppressed at this energy dose (Biomed Pharmacother 2021 Aug 16;142:112036.). Thus, we suggest that this could be an upper limit of energy dosage for biologic response of HUVECs. This is for why we did not perform more increased dosage ECSW energy for HUVECs. Additionally, different kinds of cells have different biologic effect. Our lab data showed that HUVECs has distinctive characteristics on resistant to higher energy of ECSW therapy than the other cells, such as bone marrow-derived endothelial progenitor cells (EPCs), adipose-derived mesenchymal stem cells (ADMSCs).

Comment 3: I noticed that the doses of shockwave used in Figure 1, 2, 3 (0.x mJ) were significantly lower as compared to Figure 4 onwards (1.5 mJ). What was the reason for this? Please elaborate because it could be that such dose difference would affect the interpretability of the findings.

Response 3: The anatomical structures offemoral iliac bone areas” include: skin, skeletal muscle layer and bone. Accordingly, it needed higher ECSW (1.5 mJ/mm2/200 shots) energy to penetrate not only to the skin and muscle layer, but importantly to penetrate to the bone marrow (but is the hard organ). On the other hand, it is quite different for the HUVECS which were seeded in the culture dick that was directly applied ECSW energy (i.e., need more lower ECSW energy for its biological effect).   

Comment 4: Line 354 and Figure 7B: when I see the LVEF reduction, it seems to me that although the number is lower than control, it is still within the normal range (60%). Do the authors believe that this model could sufficiently represent DCM (as claimed by the authors in line 443 and 475), which has a contractile dysfunction and most likely result in LVEF below normal range? This definitely require a justification and I am also wondering why didn't the authors impair the LVEF further with doxorubicin? I am sure it is doable. 

Response 4: Dear reviewer, first of all, we would like to sincerely thank you for your professional criticism. To your question, we have several explanations: First, to human being the normal LVEF is ≥60%; however, to rodent, the LVEF could be ≥73% to 75%. Thus, the normal range of LVEF is quite different among the human being and the rodent. Second, as you can see that for the DCM group, the LVEF <58% by day 60 after DCM induction, i.e., from 75% to 58%, reduced for nearby 20%, suggesting the DCM model was successful created for the study. We also suggest that the LVEF will still continuously reduced in DCM group if the animals to be euthanized for 6 or more than 6 months. Third, abundant data from our team demonstrated that administered intraperitoneally with an accumulated dosage of 12.5 mg/kg given to the animals at 4 separated time points within 20 days (i.e., once every 5 days) is the suitable dosage. More previously, we had also increased the dosage (i.e., accumulated to 15 mg/kg) of doxorubicin to the rodent. The result demonstrated unacceptably high complications in rodent, including the intestinal perforation, acute kidney injury, frequency of ascites, markedly reduced body weight and very high mortality rate for such a dosage of doxorubicin, especially in a cardiorenal syndrome animal model. 

Comment 5: Also, the increase with both shockwave and exogenous mitochondria showed roughly 5% increase of LVEF. Do the authors think that 5% of LVEF improvement would be clinically relevant? Would there be a significant improvement on the NYHA class?

Response 5: (1) the results showed that improvement of LVEF for ECSW or mito-ADMSC was around 6%-7%; for combined ECSW + mito-ADMSC was around 10%. (2) Sometimes, change of 5% LVEF to patients is quite important. For example: LVEF from 40% to 35% defined as from borderline to impaired LV function; LVEF from 33% to 28% defined from impaired to poor LV function. Additionally, when the patient’s LVEF ≥ 25%, he or she is not a candidate for heart transplantation. On the other hand, when the patient’s LVEF < 20%, he or she is really a candidate for heart transplantation. Consistently, we also suggest that the LVEF will still continuously reduced in DCM group but better improved in treated groups when the animals to be euthanized for 6 or more than 6 months. 

Comment 6: In the manuscript, although the authors have demonstrated extensively the effects of shockwave and exogenous mito-ADMSCs on structural remodeling, it lacks of functional investigations showing that those interventions also improve cardiomyocyte physiology, not only structures. The only functional assessment was done by measuring LVEF. However, it is not adequate since LVEF is not a cellular property. The authors could perform experiments to assess the intracellular calcium handling or calcium wave or tension, etc.

Response 6: 1) Except for the evaluation of heart function, we also deeply investigated the molecular-cellular levels of cardiomyocytes such as protein levels of α-MHC, ß-MHC, cytosolic cytochrome C, mitochondrial cytochrome C, apoptotic/fibortic biomarkers of LV myocardium, cardiomyocyte size and Masson’s trichrome stain for identification of fibrosis of LV myocardium. 2) Dear reviewer, as you know that the “intracellular calcium handling or calcium wave or tension” study is a very classical physiological study. We are honest to tell you that our lab is lacking those of instruments for example: the patch clamp technique or the other requirement tools for evaluation of Ca2+ ionic currents or voltage across the cell membrane. 3) We know that this is the limitation of our study that has been discussed on the limitation paragraph of our revised manuscript.   

Comment 7: I am not sure if ADMSC is a common abbreviation, which doesn't require any description in its first use. Therefore, adding it in the title without describing what it is would not be appropriate. Moreover, in the abstract the description of ADMSC (what it stands for) is not available. Please adapt the title so that it is well explained while also maintaining the length of the title within a reasonable limit.

Response 7: We apology for our inappropriate presentation. Yes, according to your recommendation, we have given the full name of ADMSC in the title page and in abstract of first mentioned ADMSCs in our revised manuscript.

Comment 8: Section 0 needs to be removed

Response 8: Yes, according to your recommendation, we have removed the Section 0 in our revised manuscript.

Comment 9: In general, the introduction needs to be more concise. Some words are not needed and can be removed to improve clarity of the content. 

Response 9: Yes, according to your recommendation. The inappropriate writing has been corrected in our revised manuscript.

    • Comment 9-1: "It is well recognized that the dilated cardiomyopathy (DCM), a non-ischemia related heart muscle disease with architectural..."
    • Response 9-1: Yes, this have been rewritten in our revised manuscript.
    • Comment 9-2: "The DCM is a clinical diagnosis characterized by left ventricular or biventricular dilation and impaired contraction in the absence of coronary artery disease, hypertension, valvular disease or congenital..."
    • Response 2: Yes, this has ready been corrected in our revised manuscript.
    • Comment 9-3: Line 53: Please check if "anomaly" is the most appropriate word. In my opinion, "abnormality" is more appropriate.
    • Response 9-3: “The” has been removed in our revised manuscript.
    • Comment 9-4: "Genetic inheritance arises in 30-48% of patients, and inflammatory disorders such as myocarditis or toxic effects from medications, alcohol, or illicit drugs also could result in dilated cardiomyopathy"
    • Response 9-4: These sentences have been rewritten in the Introduction paragraph of our revised manuscript.
    • Comment 9-5: Line 69: Please choose one between "dilation" and "dilatation". Check the whole manuscript and make it consistent.
    • Response 9-5: Dear reviewer: we have reviewed the literatures and found that “LV chamber dilatation” is very commonly utilized. So, we haven’t changed this word.
    • Comment 9-6: "our previous studies have revealed that combined ADMSCs and ECSW therapy was superior to either one for improving acute ischemia-related LV dysfunction"
    • Response 9-6: Yes, this phrase has been rewritten in our revised manuscript.
    • Comment 9-7: Line 88: "ADMASCs"
    • Response 9-7: Yes, this mistake has been corrected in our revised manuscript.  

Comment 10: An important remark: please make sure to remove "Figure X" from all (sub)headings and when linking the figure to the text, please specify which part of the figures we should look into. Don't say "Fig 1" only, but please use "Fig 1A" or "Fig 1B", etc.

Response 10: Yes, all of these have been corrected by you recommendation in Result section of our revised manuscript

Comment 11: Heading of section 3.2: "ECSW therapy augmented exogenous mitochondria into ADMSCs" augmented to do what? internalization? diffusion? Please clarify.

Response 11: Yes, according to your recommendation, we have added the word “internalization” into the subhead in our revised manuscript.

Comment 12: Figure 5 needs arrows as seen in other figures to point out the Sca-1 (green) and GATA4 (grey). They are not clearly visible, and readers might have difficulty confirming whether the numbers are indeed consistent with the data shown in the bar charts.

Response 12: Yes, according to your recommendation, we have added the red arrows to point out these two specific cells in our revised manuscript.

Comment 13: Line 340: "Illustrating the microscopic finding (400x) of a-smooth muscle actin (a-SMC) positive stain for identification of small vessel (i.e., diameter = 25µM) density (gray color) (red color)." I am curious, how did the authors quantify the small vessels? As far as I know, it is highly dependent on the cutting/slicing angle and therefore 2 vessels in the preparation could be from one single vessel which is cut twice in a different location?

Response 13: Dear reviewer, we exactly agree your comment. In the present study, we have tried to cut the specimen of LV myocardium in cross-section rather than in longitudinal section for avoiding cutting the same vessel in twice.

Comment 14: Not sure why there are two conclusions in this manuscript. Please merge them or remove one. 

Response 14: Yes, according to your recommendation, we have rewritten the conclusion in our revised manuscript.

Comment 15: Figure 11 needs to be linked with the text. Otherwise, this could be a nice graphical abstract. The figure also require a caption explaining what we see in the figure.

Response 15: Yes, we have discussed the Figure 11 on the Limitation of our revised manuscript. Additionally, we have also provided a caption explaining what we see in the figure in our revised manuscript.

Finally, we would like to tell your that we have had another native English speaker to edit our manuscript gain.

We would like to take this opportunity to express our appreciation for your detailed review of the article and the kindness of giving us valuable suggestions. Thank you very, very much!

Reviewer 2 Report

In this manuscript the authors present the effect of extracorporeal shock waves (ECSW)  to mitochondria in ADMSCs and its capacity in preserving would preserve LVEF in rats treated with doxorubicin.  They found that ECSW therapy enhanced exogenous mitochondria into cells both in vitro and in vivo. Their studies indicated opposite trends between LVEF and cardiomyocyte fibrosis. The study is well represented and has required controls in place.

I have some minor comments:

Page 1 Line 44-Page 2 Line 50 remove the paragraph regarding template

Figure 1 A-D scale bars are illegible, please modify

Figure 3 Scale bars missing

Author Response

Response to Reviewer's Comments (Reviewer #2)

Dear Reviewer:

Your constructive criticism is greatly appreciated. We have made the following responses to comply with your honorable suggestions (Note: The revised parts of the manuscript in response to Reviewer’s comments have been marked in pink color):

Response to reviewer’s minor comments to author

Comment 1: Page 1 Line 44-Page 2 Line 50 remove the paragraph regarding template

Response 1: Yes, this has been corrected in our revised manuscript.

Comment 2: Figure 1 A-D scale bars are illegible, please modify

Response 2: We apology for this inconvenience reading. For resolving this problem, we have added the explanation, including the scale bar and the manifestation of microscope in the Figure legend of our revised manuscript. 

Comment 3: Figure 3 Scale bars missing

Response 3: Dear reviewer, we are honest to tell you that the Scale bars in A1 to D4 were not missing but it was too small, resulting in difficult to be read. However, when we zoom the picture larger and larger, we can really find the Scale bar. Because this illustration is immunofluorescent staining and the examination slice had been completed for a rather long interval, resulting in the immunofluorescent color is dedicated. Therefore, we are sorry for that we cannot provide the new picture with clear Scale Bar. To resolve this problem, we have added the scale bar and the manifestation of microscope in the Figure legend of our revised manuscript for readers to understand these values.  

 Finally, we would like to tell your that we have had another native English speaker to edit our manuscript gain.

We are greatly indebted to you for your professional comments.

Round 2

Reviewer 1 Report

Thank you for addressing my previous comments. I have no further remarks, except for some typos and errors that need corrections:

  • line 36: "cytochroem" should be "cytochrome"?
  • I don't see any "red arrows" in Figure 5. Please add.
  • Line 518: Please remove this part of the sentence in the figure legend as it is not necessary: "Figure 11 summarized that".
  • Please make sure that "Ca2+" is written as "Ca2+".

Author Response

Response to Comments and Suggestions for Authors

Note: The revised parts of the manuscript in response to Reviewer’s comments have been marked in red color)

Comment 1: line 36: "cytochroem" should be "cytochrome"?

Response 1: We apology for our typo error. This has been corrected in our revised manuscript. 

Comment 2: I don't see any "red arrows" in Figure 5. Please add.

Response 2: Yes, we have submitted the correction in Figure 5 in our revised manuscript. 

Comment 3: Line 518: Please remove this part of the sentence in the figure legend as it is not necessary: "Figure 11 summarized that".

Response 3: Yes, according to your recommendation, we have deleted these sentences in our revised manuscript

Comment 4: Please make sure that "Ca2+" is written as "Ca2+".

Response 4: Yes, this inappropriate writing has been corrected in our revised manuscript.

We would sincerely like to thank you again for your kindly help!

Round 3

Reviewer 1 Report

I have no further remarks, except for this one:

"Figure 11. summarized that [ADD a SHORT TITLE HERE]. ECSW-facilitated exogenous mitochondria into ADMSCs therapy ameliorated the LV function mainly through enhancing angiogenesis, expression of cardiac stem cells/growth factors in LV myocardium, mobilization of bone marrow-derived endothelial progenitor cells into circulation, MSC differentiation and soluble angiogenesis factors. ADMSCs = adipose-derived mesenchymal stem cells; ECSW = extracorporeal shock wave; DCM = dilated cardiomyopathy; LVEF = left ventricular ejection fraction; CSC = cardiac stem cell; VEGF = vascular endothelial growth factor; TGF-ß = transforming growth factor beta."